# The Effect of Extraction Methods on Phytochemicals and Biological Activities of Green Coffee Beans Extracts

**DOI:** 10.3390/plants12040712

**Published:** 2023-02-06

**Authors:** Octavia Gligor, Simona Clichici, Remus Moldovan, Dana Muntean, Ana-Maria Vlase, George Cosmin Nadăș, Ioana Adriana Matei, Gabriela Adriana Filip, Laurian Vlase, Gianina Crișan

**Affiliations:** 1Department of Pharmaceutical Botany, Iuliu Hațieganu University of Medicine and Pharmacy, 8 Victor Babeș Street, 400347 Cluj-Napoca, Romania; 2Department of Physiology, Iuliu Hațieganu University of Medicine and Pharmacy, 8 Victor Babeș Street, 400347 Cluj-Napoca, Romania; 3Department of Pharmaceutical Technology and Biopharmaceutics, University of Medicine and Pharmacy, 8 Victor Babeș Street, 400347 Cluj-Napoca, Romania; 4Department of Microbiology, University of Agricultural Sciences and Veterinary Medicine, 3/5 Mănăștur Street, 400372 Cluj-Napoca, Romania

**Keywords:** green coffee beans, innovative extraction methods, oxidative stress reduction, antimicrobial activity, anti-inflammatory activity, ultrasound-assisted extraction

## Abstract

The objectives of the present study consisted of identifying the impact of extraction methods and parameters held over the phytochemistry and biological activities of green coffee beans. Extraction processes belonging to two categories were performed: classical methods—maceration, Soxhlet extraction, and such innovative methods as turboextraction, ultrasound-assisted extraction, and a combination of the latter two. Total polyphenolic and flavonoid content, as well as in vitro antioxidant activity of the resulted extracts were spectrophotometrically determined. Extracts displaying the highest yields of bioactive compounds were subjected to High Performance Liquid Chromatography-Mass Spectrometry analysis. The extracts with the best phytochemical profiles were selected for biological activity assessment. In vivo, a model of plantar inflammation in Wistar rats was used to determine antioxidant activity, by evaluating the oxidative stress reduction potential, and anti-inflammatory activity. In vitro antimicrobial activity was also determined. The Soxhlet extraction and ultrasound-assisted extraction gave the highest bioactive compound yields. The highest total polyphenolic content was 2.691 mg/mL gallic acid equivalents and total flavonoid content was 0.487 mM quercetin equivalents for the Soxhlet extract subjected to 60 min extraction time. Regarding the antioxidant activity, ultrasound-assisted extraction reached the highest levels, i.e., 9.160 mg/mL Trolox equivalents in the DPPH (2,2-diphenyl-1-picryl-hydrazyl-hydrate) assay and 26.676 mM Trolox equivalents in the FRAP (Ferric Reducing Antioxidant Power) assay, at a 30 min extraction time and 50 °C extraction temperature. The 60 min Soxhlet extract reached the highest level for the ABTS^+^ (2,2′-azino-bis(3-ethylbenzothiazoline-6-sulfonic acid)) assay, 16.136 mM Trolox equivalents, respectively. Chlorogenic acid was present in the highest concentration in the same Soxhlet extract, 1657.179 µg/mL extract, respectively. Sterolic compounds were found in high concentrations throughout all the analyzed extracts. A proportional increase between yields and extraction parameter values was observed. Increased inhibition of Gram-negative bacteria was observed. The finally selected Soxhlet extract, that of 60 min extraction time, presented a significant in vivo antioxidant activity, with a slight anti-inflammatory activity. Antioxidant levels were elevated after 2 h of extract administration. Pro-inflammatory cytokine secretion was not influenced by the administration of the extract.

## 1. Introduction

Coffee beans refer to the seeds of the coffee plant (the genus *Coffea* L.), botanically considered as a shrub or tree. The coffee plant is part of the botanical family Rubiaceae. Presently, the family includes hundreds of plant species, with the genus *Coffea* encompassing over 70 distinct species. Two of these species, present economic importance—*Coffea arabica* L. (often named Arabica) and *Coffea canephora* Pierre ex A. Froehner (often named Robusta). This article focuses on the species that was first mentioned, *Coffea arabica* L. The evergreen plant grows up to 10 m in height, consisting of a main stem, originating secondary branches, with leaves being disposed in an opposite decussate arrangement and colored dark green. The seeds of the plant are located inside the fruits, named cherries or berries. The seeds are elliptical in shape. [1,2,3]. The process of roasting, although holding great industrial importance, due to its implication in the palatability of the final beverage, has been noted to cause the degradation of numerous bioactive compounds beneficial for human health, such as polyphenolic compounds, polysaccharides, proteins, and many others. This matter subsequently reduces the biological potential of the plant material [4,5,6].

Green coffee beans have been reported to present numerous biological activities, such as obesity reduction, type II diabetes prevention, reducing oxidative stress, improving cardiovascular parameters, reducing the risk of chronic hepatic disorders, and also neuroprotective, antitumor, antioxidant, anti-inflammatory, and antimicrobial effects [7,8,9,10]. This wide array of biological activities is mainly considered to stem from the polyphenolic compounds found in this plant material [2].

In order to ensure the availability of such bioactive compounds, along with offering high recovery rates from any plant matrix, the extraction process holds much importance. Generally, extraction methods are divided into two main categories, based on the era in which they appeared. Names such as “conventional” or “classical” methods are attributed to the methods used prior to the end of the 20th century. Whereas the terms “innovative” or “emerging” extraction methods, are used for those methods that appeared thereafter. Conventional methods entail high temperatures, long periods of time in order for the extraction process to reach completion, large solvent volumes, large amounts of plant material, as well as often hazardous solvents. Maceration, decoction, Soxhlet extraction constitute examples of such conventional methods. These characteristics are presently regarded as unfavorable, thus urging for safer, less time-consuming, alternative means of bioactive compound extraction methods to be discovered, with lower amounts of alternative solvents and plant matrix also being required. Thus, bearing the name “green extraction methods” due to their indicative safety, microwave-assisted extraction, ultrasound-assisted extraction, pressurized liquid extraction, etc. are considered clear examples of such methods [11,12].

While several reports on green coffee beans provide outlooks over the influence of extraction method and parameters over bioactive compounds yields, few have focused on the importance of such aspects of the extraction process over the biological activities of the resulted extracts [13,14,15].

The purpose of this article was to provide a better understanding of the relationship between extraction type and extraction conditions and the final extraction yields, as well as to determine the influence such extraction parameters might hold over the biological properties of the obtained green coffee bean extracts, such as antimicrobial activity, antioxidant, and anti-inflammatory effects.

## 2. Results

Sample nomenclature was based on extraction method and the parameters that were studied for each sample: M for maceration, S for Soxhlet extraction (SE), U for Ultrasound-assisted extraction (UAE), T for turboextraction (TBE), and UT for the combination for the UAE and TBE (UTE). A number of 20 different samples resulted after extraction. One sample was obtained after maceration (M), 3 samples were obtained through SE (S), 6 samples were obtained by TBE (T), 9 samples were obtained by UAE (U), and the last sample was obtained by UTE (UT). The nomenclature of the employed extraction methods, as well as that of the obtained samples is clarified in Table 1.

Total polyphenolic content (TPC) and total flavonoid content (TFC), as well as antioxidant capacity were evaluated for the samples listed above, following the methods detailed in chapter 4. Materials and Methods. Chromatographic evaluation followed only for the samples which presented the highest values for the analysis processes previously stated. The final step of the study consisted of the biological activity assessment of the samples containing the highest yields of bioactive compounds.

### 2.1. Influence of Extraction Parameters on TPC and TFC values

Table 2 depicts the results of TPC and TFC assessment of the green coffee bean samples. As seen in the respective table, the SE method attained the highest content of polyphenols, with the 60 min extraction period being superior to all samples. SE values were followed by maceration and TBE in regard to polyphenolic yields. UAE and UTE achieved the lowest levels of yields for both assessments.

### 2.2. Influence of Extraction Parameters on In Vitro Antioxidant Capacity

Results for in vitro antioxidant capacity assessment are detailed in Table 3. Following the DPPH assay, no major differences were observed throughout the samples, excepting the UAE sample with the extraction conditions of 30 min extraction time and 50 °C extraction temperature, i.e., sample U35. This value was closely followed by that of the extract obtained through the combination of UAE and TBE, i.e., sample UT.

For the FRAP assay, the same UAE sample, U35, showed the highest value. However, notable differences between the rest of the extraction methods were observed, namely SE, i.e., samples S20 and S60, as well as UAE, i.e., samples U15 and U23, which reached medium to high levels. For UAE, in this case, there was a visible decrease in levels for samples that were subjected to higher time periods and temperature values. For TBE, an increase in antioxidant levels could be observed, along with the increase in the values of the extraction parameters, up until the highest of these values were reached, namely the parameters of 4 cycles of 5 min (20 min) with 6000 rpm speed, i.e., sample T46. After which, an abrupt decrease in the antioxidant levels in this case, for sample T48. This extract was obtained at the same extraction time but at 8000 rpm.

Results were evidently different for the ABTS^+^ assay. In this case, the Soxhlet sample with the 60 min extraction period, S60, as well as all TBE samples (of both extraction times, 10 min, and 20 min, respectively) reached the highest levels.

### 2.3. HPLC-MS Analysis of the Extracts

Following the assessments of the previous subchapters, Section 2.1. Influence of extraction parameters on TPC and TFC values, and Section 2.2. In vitro antioxidant capacity of the extracts, 11 extracts were selected for further analysis. The main criterion for selection was the highest yield level in the extracts. Thus, the samples were screened for polyphenolic compounds, flavonoid compounds, and sterolic compounds (Table 4).

#### 2.3.1. Analysis of Polyphenolic and Flavonoid Compounds

Only one polyphenolic compound, chlorogenic acid, was identified in the samples that were selected for chromatographic evaluation. The highest yield of chlorogenic acid was recorded for the SE sample with extraction time of 60 min, sample S60. Other notable yields were obtained also through SE, with extraction time of 20 min and 40 min, samples S20 and S40, respectively. TBE samples reached comparable yields, namely the conditions of 4 cycles of 5 min and rotation speed of 4000 rpm, and 6000 rpm, samples T44 and T46, respectively. Maceration, UAE, and UTE reached only low to medium yields of chlorogenic acid.

Additionally, kaempferol was the only flavonoid compound identified in one of the samples. The extract produced through maceration, sample M, was the only extract in which kaempferol reached quantifiable levels.

#### 2.3.2. Analysis of Sterolic Compounds

6 sterolic compounds were found in high levels in the majority of the samples: α-tocopherol, γ-tocopherol, ergosterol, stigmasterol, β-sitosterol, and campesterol. Results are presented in Table 4.

UAE was the only method that managed the extraction of α-tocopherol. All 3 UAE samples along with the sample resulted from UTE, sample UT, presented nearly equal values of α-tocopherol.

γ-Tocopherol was identified in all UAE samples albeit in higher levels, depending on the extraction parameters. As in the case for its previously discussed isomer, UAE and UTE enabled the highest yields. For UAE, the extraction conditions of 30 min and 50 °C proved most advantageous, i.e., sample U35.

Ergosterol was detected only partially in the samples, most notably, the S20 sample, followed by TBE samples, especially for the parameters of 4 cycles of 5 min and 4000 rpm, i.e., sample T44. The latter was followed closely by UAE (specifically 30 min, 50 °C—sample U35) and the UT sample.

High levels of stigmasterol were also found in most of the samples, i.e., TBE (2 cycles of 5 min, and 4000 rpm—sample T24), UAE (30 min, 50 °C—sample U35), followed closely by maceration and SE (60 min—sample S60).

β-Sitosterol, the sterolic compound present in the highest levels in the analyzed samples, was also better extracted by TBE (2 cycles of 5 min, 4000 rpm—sample T24) and UAE (30 min, 40 °C —sample U34), followed by SE (60 min extraction time—S60) and UTE.

Campesterol was found in high levels in the 60 min SE sample (S60), followed by a TBE sample, i.e., 2 cycles of 5 min and 4000 rpm (T24), and a UAE sample, i.e., 30 min and 50 °C (U35).

### 2.4. Determination of Antimicrobial Activity

Following the assessment of the phytochemical profile by HPLC-MS, the extracts that were considered to present the most favorable profiles were further subjected to antimicrobial activity determination. Therefore, extracts S60 and U35 were selected for this evaluation.

#### 2.4.1. In Vitro Qualitative Study of Antimicrobial Activity

The disk diffusion test was applied in order to assess the antimicrobial potential of the samples. The samples presented an increased efficiency against Gram-negative bacteria, a moderate efficiency against Gram-positive bacteria, and reduced activity towards *Candida albicans*. Values are shown in Table 5. Diameters of inhibition areas are as follows: 5.86 to 7.29 mm for Gram-positive species, 10.09 to 13.38 mm for Gram-negative bacteria, and 6.94 to 7.21 mm for *Candida albicans*. Results demonstrated an increased antimicrobial activity against Gram-negative bacteria.

#### 2.4.2. In Vitro Quantitative Study of Antimicrobial Activity

The potential against Gram-negative strains of the samples was evidenced by the first screening method. However, the MIC (minimum inhibitory concentration) method was utilized in order to assess the quantitative antimicrobial potential against all microbial species in the initial qualitative study. Table 6 presents the varied antimicrobial response of the extracts.

### 2.5. Assessment of Oxidative Stress and Inflammation Markers

Extract S60 was selected for in vivo biological activity evaluation. As this extract was administered to the experimental animals solely against control groups receiving anti-inflammatory treatment (Indomethacin) or carboxymethylcellulose (CMC), the authors would like to note that extract S60 is to be referred simply by its extraction method name, that is SE, for the remainder of this chapter (S60 being the only extract obtained SE that was further studied).

Lipid peroxidation marker (MDA) and endogenous antioxidant levels such as reduced glutathione (GSH), oxidated glutathione (GSSG), and their respective ratio (GSH/GSSG) were assessed in order to quantify the oxidative stress. Antioxidant enzymes were also measured by catalase (CAT) and glutathione peroxidase (GPx) activities in the plantar tissues. Results are displayed in Figure 1. MDA levels remained elevated at 2 h and 24 h for the SE treated group compared to control group while GSH levels increased significantly after 24 h compared to control and Indomethacin treated groups (*p* < 0.05 and *p* < 0.01). MDA levels decreased only in animals treated with Indomethacin, both at 2 h and 24 h compared to control (*p* < 0.01 and *p* < 0.001). GSSG levels diminished after SE treatment, at 2 h, compared to control group (*p* < 0.01), similar to the values recorded after Indomethacin treatment (*p* < 0.001). The effect of SE on GSSG formation was more pronounced at 24 h (*p* < 0.001) than at 2 h. Another noteworthy observation was the increase in the GSH/GSSG ratio at 2 h for Indomethacin group vs. control (*p* < 0.05). CAT activity was amplified significantly in group treated with Indomethacin, at 24 h after induction of inflammation, compared to control group while GPx activity decreased significantly after Indomethacin administration, both at 2 h (*p* < 0.05) and 24 h (*p* < 0.05). In SE group, GPx activity was comparable to the control group, both at 2 h and 24 h (*p* > 0.05).

The levels of pro-inflammatory cytokines, i.e., IL-6 and TNF-α, were also evaluated in the plantar tissue, at 2 h and 24 h after induction of inflammation. Figure 2 portrays the obtained results. Thus, SE reduced IL-6 secretion in the paw tissue, at 2 h compared to control, but the results were statistically insignificant (*p* > 0.05). The IL-6 levels diminished significantly after Indomethacin administration (*p* < 0.05). SE treatment did not influence the IL-6 secretion in soft plantar tissue at 24 h after carrageenan injection (*p* > 0.05). TNF-α levels in the paw tissue, measured at 2 h after induction of inflammation, decreased only in Indomethacin group (*p* < 0.05) while SE maintained high levels of this cytokine, close to control group (*p* > 0.05) At 24 h, the treatments did not induce significant difference between groups in TNF-α secretion, in the soft plantar tissue.

## 3. Discussion

The seeds of *Coffea arabica* L. comprise the majority of the worldwide production of coffee. They are situated within the fruits of the plant, which are also named cherries or berries. The fruit consists of an exocarp (skin of the fruit), a mesocarp containing pectins, an endocarp containing polysaccharides, and the silver skin, which coats the seed, containing polysaccharides, proteins, polyphenols, etc. A physical characteristic of the seeds is their elliptical shape [1,3]. Once harvested, the seeds are subjected to dehulling and processing by dry or wet methods, followed by roasting, which is responsible for conferring the aroma of the finally obtained coffee drink. However, roasting has been noted to cause the degradation of polysaccharides, lipids, chlorogenic acids, other polyphenolic compounds, trigonelline (a compound also responsible for flavor), and protein denaturation. It has been observed that these changes led to reduction in the subsequent biological activities of the plant material [4,5,6].

Pimpley et al. reviewed the effect of roasting of coffee beans over the content of chlorogenic acids (among which hydroxycinnamic acids are a part of, such as caffeic acid, ferulic acid and p-coumaric acid). The study concluded that up to 95% of said compounds were degraded through intense roasting, leaving a very small percentage of these compounds in the plant material. In addition, coffee beans and pulp extracts were reported to inhibit lipid accumulation in cell cultures of adipocytes. Additionally, green coffee beans extracts were observed to reduce obesity and insulin resistance in mouse models. In human test subjects, a decaffeinated green coffee bean extract led to decreased metabolic syndrome markers, such as lipid profile, blood pressure, insulin resistance, etc. [7]. Martínez-López et al. reported that a moderate consumption of a green/roasted (35:65) blend improved cardiovascular parameters in human subjects with moderate hypercholesterolemia, such as serum lipid profile, blood pressure, body weight, while also increasing plasma antioxidant capacity [8]. Caro-Gómez et al. also noted ameliorated cardiometabolic syndrome parameters, such as fasting glucose, insulin resistance, liver triglyceride levels, as well as increasing IL-6 levels and positively impacting gut microbiota in ApoE^−/−^ mice following an administration of green coffee beans extract [9]. A study conducted by Wang et al. on endothelial EA.hy926 cells pretreated with green coffee beans polyphenolic extracts, demonstrated endothelial protective effects by reducing the production of reactive oxygen species, and increasing endothelial nitric oxide synthase levels [10].

Recent scientific articles have investigated the influence of extraction methods over the biochemical profile of green coffee beans extracts. Yuniarti et al., for example, have optimized an extraction process assisted by natural deep eutectic solvents for caffeine and chlorogenic acid from the green beans pertaining to the species *C. canephora* L. Optimal conditions were found to be 4:1 for the mole ratio of choline chloride-sorbitol, 60 min of extraction time, and 1:30 g/mL for the liquid-solid ratio. The extraction method used as reference was maceration [13]. Menzio et al. managed to enhance mass transfer and selectivity of caffeine extraction from green coffee beans by combining supercritical CO_2_ extraction with UAE [14]. Gawlik-Dziki et al., apart from demonstrating the in vitro capacity of lowering lipoxygenase levels of methanolic green coffee beans extracts, also compared plant materials from different locations worldwide [15].

However, it is the authors’ opinion that further research must be conducted on the correlation between the extraction methods as well as extraction parameters and biological activity, particularly for this plant material.

A tendency of increase in TPC was observed along with the increase in extraction time, for each of the extraction methods employed in this study. However, in the case of UAE, an increase in TPC was observed along with the increase in extraction temperature, as well. A similar report, although for a different version of plant material, spent filter coffee, by Pavlović et al. stated that the increase in extraction time led to a decrease in TPC, a finding not applicable in the case of this study [16].

Concerning the TFC of the samples, minor differences were observed throughout the majority of the samples, regardless of the extraction technique used. The sole exception to this observation was once again the Soxhlet extraction, namely the sample obtained after 60 min extraction time (sample S60). Al-Dhabi et al. have reported the positive influence that temperature and extraction time held over waste spent coffee grounds subjected to a UAE method. However, a further increase in these parameters was observed to obtain lower TPC and TFC values, due to the degradation of the compounds as a result of excessive exposure [17]. This observation was not found in this study, as extraction time for UAE did not exceed 30 min, and the highest temperature value was 50 °C. A previously cited study, led by Pavlović et al., also observed an increase in antioxidant capacity for DPPH and FRAP assays, with the decrease in extraction time, for the case of a microwave-assisted extraction for spent filter coffee. In addition, a proportional correlation between TPC values and antioxidant capacity (in the case of DPPH and FRAP assays) was observed for the previously mentioned study [16]. As seen in Table 3, no notable correlation between results for the employed assays could be observed within this study. At the very least, sample S60, which presented the highest TPC and TFC levels, registered the highest value for the ABTS^+^ assay as well.

Contrary to the present HPLC findings, Nzekoue et al. have reported most of the quantified compounds mentioned above (see Table 4), for hydroalcoholic and hydromethanolic sonicated extracts, albeit the plant material of that study consisted of roasted coffee silver skin. Which is known to contain multiple unconjugated polyphenolic compounds, as opposed to green coffee beans [18]. Results for sterolic compounds were in accordance to previously reported data by Dong et al. for green coffee oil extracted by a UAE method [19]. Although an increase in extraction parameter values led to a somewhat improvement in concentrations for stigmasterol, β-sitosterol, and campesterol, the results were contrary for TBE. In that case, the extraction time of 4 cycles of 5 min (20 min) and 6000 rpm could have possibly contributed to the degradation of the compounds, see sample T46. The lowest levels for stigmasterol, β-sitosterol, and campesterol were also registered in UAE samples, namely, the conditions of 20 min extraction time and 30 °C (sample U23), which might indicate these extraction parameters as insufficient.

Concerning the antimicrobial activity of the analyzed samples (samples U35 and S60), Nzekoue et al. have contrarily reported a low activity of similarly achieved extracts of roasted coffee silver skin [18]. Another noteworthy observation of the present study would consist in that of the lower MICs registered for the Gram-negative species. A possible reason for this phenomenon could be the limited diffusion on the agar surface as opposed to an appropriate overall inhibitory concentration in wells containing liquid MH medium.

In terms of in vivo activity, the oxidative stress reduction potential of the analyzed extract, SE, could be a result of the present sterolic compounds which were reported to have antioxidant activity [20]. The results confirmed this finding by a good in vivo antioxidant effect of the SE extract, especially at 2 h after induction of inflammation. SE extract increased the reduced glutathione level and diminished the oxidation of glutathione in the plantar tissue, the effect having lasted even 24 h. The present results were comparable with those of recent scientific literature. Bhandarkar et al. have found that green coffee beans only ameliorated heart and liver inflammation in Wistar rats fed with high-carbohydrate, high-fat diet with green coffee extract (5% in food), with or without caffeine, compared to control groups. Results were correlated with those of rats which received a corn starch diet or a high-carbohydrate, high-fat diet [21]. Pergolizzi et al. found that the application of a *C. robusta* L. Linden ointment induced a prolonged and sustained anti-inflammatory effect on carrageenan-induced rat oedema [22]. In human subjects, Martínez-López et al. noticed a tendency towards the reduction in pro-inflammatory cytokines, among which IL-6 and TNF-α, associated with green coffee consumption by human subjects, albeit not significantly. However, MDA levels decreased after human coffee consumption [8]. According to the results of a meta-analysis of randomized clinical trials, performed by Asbaghi et al., such effects of green coffee beans are due to their phenolic compounds, i.e., caffeoylquinic acids. These compounds have been observed to reduce pro-inflammatory cytokines in both liver and white adipose tissue in humans, especially lowering IL-6 and TNF-α levels [23].

## 4. Materials and Methods

### 4.1. Plant Material, Reagents, and Laboratory Equipment

Ground green coffee beans (*Coffea arabica* L.) were procured from a local company (All For Nature, Timișoara, Timiș, Romania).

Folin-Ciocâlteu reagent, sodium carbonate (Na_2_CO_3_), Aluminum chloride (AlCl_3_), ABTS^+^ (diammonium 2,2′-azino-bis(3-ethylbenzothiazoline-6-sulfonate), DPPH (2,2-Diphenyl-1-(2,4,6-trinitrophenyl)hydrazyl), TPTZ (2,4,6-Tris(2-pyridyl)-s-triazine), indomethacin, carboxymethylcellulose, o-phthalaldehyde, Lambda carrageenan type IV were purchased from Sigma–Aldrich (Taufkirchen, Germany). 2-thiobarbituric acid and Bradford reagent were acquired from Merck KGaA (Darmstadt, Germany) and ELISA cytokines tests (TNF-α and IL-6, respectively) were obtained from Elabscience (Houston, TX, USA). Bradford total protein assay was purchased from Biorad (Hercules, CA, USA). All analytical grade, HPLC reagents and standards were acquired from Sigma–Aldrich (Taufkirchen, Germany) and Decorias (Rediu, Romania).

The following equipment was used for the present study: SER 148 solvent extraction unit (VELP^®^ Scientifica, Usmate Velate, Italy), T 50 ULTRA-TURRAX^®^ disperser (IKA^®^-Werke GmbH & Co. KG, Staufen, Germany), Sonic-3 ultrasonic bath (Polisonic, Warsaw, Poland), refrigerated high speed centrifuge Sigma 3-30KS (Sigma Laborzentrifugen GmbH, Osterode am Harz, Germany), Specord 200 Plus spectrophotometer (Analytik Jena, Jena, Germany), Agilent 1100 Series HPLC Value System coupled with an Agilent 1100 mass spectrometer (LC/MSD Ion Trap SL) (Agilent Technologies, Santa Clara, CA, USA), Bioblock Scientific 94200 rotary evaporator (Heidolph Instruments GmbH & Co. KG, Schwabach, Germany), vacuum controller HS-0245 (Hahnshin Scientific Co., Tongjin-eup, Gimpo-si, Gyeonggi-do, South Korea), Brinkman Polytron homogenizer (Kinematica AG, Littau-Luzern, Switzerland).

### 4.2. Extraction Methods

70% ethanol was used as solvent, with the selected solvent to sample ratio being 10:1 (v/w). These parameters were chosen to remain constant throughout all extraction processes in order to allow results uniformity and to provide coherence for the comparison between extraction methods. Once each extraction process was finished, samples were separated by centrifugation at 12000 rpm, for 10 min.

#### 4.2.1. Maceration

The procedure for this extraction method followed the conditions provided by the Romanian Pharmacopoeia. Respectively, 50 mL 70% alcohol were added to 5 g plant material, in a Falcon flask. The mixture was left for a period of 10 days, at room temperature, and submitted to periodical agitation. Following extraction, the sample was centrifugated in order to ensure separation.

#### 4.2.2. Soxhlet Extraction (SE)

For each sample, 5 g plant material was added in an extraction cup, along with 50 mL 70% alcohol. The heating plate temperature was set to 210 °C and the studied extraction time values were: 20 min, 40 min, and 60 min. The samples were separated once the extraction process was completed.

#### 4.2.3. Turboextraction (TBE)

The parameters studied for this extraction process were time and rotation speed. Extraction time was divided into 2 cycles of 5 min (a total of 10 min), and 4 cycles of 5 min (a total of 20 min), respectively. The studied rotation speed values were 4000, 6000, and 8000 rpm. This manner of experimentation was considered to be advantageous as it limited the risk of solvent evaporation and device overheating. The samples were centrifugated afterwards.

#### 4.2.4. Ultrasound-Assisted Extraction (UAE)

The studied extraction time values were 10, 20, and 30 min. The assessed temperature values were 30°, 40°, and 50 °C. The constant parameters were frequency, 50 Hz, and power, 230 V, respectively. The samples were separated following extraction.

#### 4.2.5. Combination of UAE and TBE (UTE)

In order to provide an efficient combination of these two extraction methods, the selected parameters remained fixed throughout the process. Thus, the ultrasonic bath was brought to 30 °C, the disperser speed was set to 4000 rpm, and extraction time was reduced to 5 min. Fixed values were selected in this case so as to prevent solvent evaporation and overheating of the two devices.

### 4.3. Assessment of Total Phenolic Content (TPC)

In order to determine the total polyphenolic content, the Folin-Ciocâlteu method was applied, following recommendations provided Csepregi et al., with several modifications [24]. 270 µL Folin-Ciocâlteu reagent were added to 60 µL plant extract, followed by 270 µL 6% Na_2_CO_3_ (*w*/*v*). After 30 min incubation, in an environment devoid of light, sample absorbances were determined at 765 nm. Gallic acid was selected as standard, and results were therefore expressed as mg gallic acid equivalents per mL (GAE mg/mL).

### 4.4. Assessment of Total Flavonoid Content (TFC)

An adapted version of the method employed by Pinacho et al. was used to determine the flavonoid content of the samples [25]. As such, 400 µL solution of AlCl_3_ 20 mg/mL in 5% acetic acid in ethanol 3:1 (*v*/*v*) ratio were mixed with 200 µL plant extract. Measurements were carried out at 420 nm wavelength. Quercetin was selected as standard. Finally, results were given as mM quercetin equivalents (QE mM).

### 4.5. In Vitro Antioxidant Capacity

#### 4.5.1. DPPH Radical Scavenging Activity

One of the methods used to evaluate the antioxidant potential of the samples was the DPPH assay. This experiment was carried out following the indications of Martins et al., after several adaptations [26]. 200 µL extract was mixed with 800 µL DPPH radical methanolic solution. Following an incubation of 30 min, in a dark environment, at 40 °C, the absorbances of the samples were measured at 517 nm. Trolox reagent was selected as standard. Results were expressed as mg Trolox equivalents per mL extract (TE mg/mL).

#### 4.5.2. ABTS^+^ Scavenging Activity

This assay was performed according to a method used by Erel et al. [27]. 200 µL acetate buffer 0.4 M, pH 5.8 were added to 20 µL ABTS^+^ in acetate buffer 30 mM, pH 3.6, with the addition of 12.5 µL extract to the mixture obtained earlier. Absorbances were measured at 660 nm. Trolox reagent was selected as standard, and results were therefore expressed as mM Trolox equivalents (mM TE).

#### 4.5.3. FRAP Assay

Experimentation was performed according to Csepregi et al. [24], i.e., FRAP reagent was obtained by adding 25 mL acetate buffer (300 mM, pH 3.6) to 2.5 mL TPTZ solution (10 mM TPTZ in 40 mM HCl) and 2.5 mL FeCl_3_ (20 mM in water). The newly prepared reagent was mixed with 30 µL extract. The mixture was incubated for 30 min, after which absorbances were determined at 620 nm. Trolox reagent was chosen as standard. Results were given as mM Trolox equivalents (TE mM).

### 4.6. Chromatographic Analysis

The phytochemical profile of the extracts was investigated by liquid chromatography tandem mass spectrometry (LC-MS/MS) with two distinct analytical methods, previously validated [28,29]. The following equipment was used: Agilent Technologies 1100 HPLC Series system (Agilent, Santa Clara, CA, USA) equipped with column thermostat, auto sampler type G1313A, binary gradient pump type G13311A, degasser type G1322A, and UV detector type G1316A. A mass spectrometer from Agilent was coupled with this system (MS with Ion Trap 1100 SL (LC/MSD Ion Trap VL, Agilent, Santa Clara, CA, USA).

The first analytical method was slightly modified and was applied to identify and quantify 23 polyphenols in vegetal extracts [28,29,30,31]. Chromatographic separation was performed on a reverse phase analytical column (Zorbax SB-C18, 100 mm × 3.0 mm id, 3.5 μm, Agilent Technologies, Santa Clara, CA, USA) with a mobile phase consisting of a mixture of methanol: acetic acid 0,1% (*v*/*v*) and a binary gradient. Elution began with a linear gradient, initially with 5% methanol and ending with 42% methanol at 35 min. For the next 3 min, isocratic elution followed with 42% methanol. Further, the column was rebalanced with 5% methanol for the following 7 min, as previously detailed [28,29,30,31]. Afterwards, the bioactive compounds were detected in both UV and MS mode. For detection of polyphenolic acids, the UV detector operated at λ = 330 nm (up to 17 min). Afterwards, for detection of the flavonoids and their aglycones, the UV detector operated at λ = 370 nm (up to 38 min). The MS system operated using an electrospray ionization source (ESI) in negative mode [28,29,30,31].

The second LC-MS analytical method was used to identify other 6 polyphenols from plant samples, as detailed in previous studies [32]. The same equipment and analytical column as aforementioned were used for chromatographic separation. The mobile phase consisted in a mixture of methanol: acetic acid 0.1% (*v*/*v*) and a binary gradient. Briefly, at start—3% methanol; at 3 min—8% methanol; from 8.5 min to 10 min—20% methanol. The column was rebalanced with 3% methanol [32]. Bioactive compounds were detected in plant samples in MS mode with the MS system operating with an ESI in negative mode.

For identification of each bioactive compound from the plant extracts, spectra from library were compared with the MS spectra/traces. To quantify the compounds, after MS detection, a UV trace was used. For the identified compounds, the calibration curve of their corresponding standards was considered for quantification of their peak areas [28,29,30,31,32].

The phytosterols were determined according to a previously validated LC-UV-MS/MS method [33,34,35]. The same equipment and chromatographic analytical column were used. However, the elution of the compounds was performed in an isocratic manner. The mobile phase consisted in a mixture of acetonitrile: methanol (90:10, *v*/*v*), a flow rate of 1 mL/min at 45 °C and 5 μL injection volume. The same mass spectrometer was used, equipped with an ion trap and atmospheric-pressure chemical ionization (APCI) source, operating in positive ionization mode. The working conditions were carefully adjusted to reach maximum sensitivity [33,34,35]. Five external standards were used for complete identification of the compounds, which was performed by comparing the retention times and mass spectra. To reduce interference and for detection of bioactive compounds, multiple reaction monitoring mode (MRM) was employed.

The Agilent ChemStation (vB01.03) and the DataAnalysis (v5.3) software were used for the acquisition and investigation of chromatographic data. All results were expressed as micrograms of bioactive compound per mL (μg/mL) of vegetal extract.

The Appendix A contains the UV chromatograms of the analyzed samples (Appendix A) as well as the analytical parameters of the database (Appendix A).

### 4.7. Antimicrobial Activity Evaluation

#### 4.7.1. In Vitro Qualitative Study of Antimicrobial Activity

The antimicrobial potential of the samples was evaluated by means of the disk diffusion method against standards consisting of strains of Gram-positive, Gram-negative bacteria, and yeasts. The following Gram-positive strains were selected as standards: *Staphylococcus aureus* ATCC 6538P, *Listeria monocytogenes* ATCC 13932, *Enterococcus faecalis* ATCC 29212, and *Bacillus cereus* ATCC 11778. Gram-negative strains standards consisted of *Escherichia coli* ATCC 10536, *Salmonella enteritidis* ATCC 13076 and *Pseudomonas aeruginosa* ATCC 27853. *Candida albicans* ATCC 10231 was selected as a yeast strain standard. Standard antibacterial and antifungal controls were amoxicillin, for bacteria and ketoconazole for the yeast. Screening was carried out according to EUCAST standards [36].

#### 4.7.2. In Vitro Quantitative Study of Antimicrobial Activity

The quantitative evaluation was performed by means of the minimum inhibitory concentration (MIC) method for the same eight standard microbial strands. The method was performed in accordance with the EUCAST protocols [36], with slight modifications. 96-wells titer plates, containing the extracts diluted in liquid MH medium and inoculated with 20 µL microbial suspension, were used. Extract stock solutions were diluted using a two-fold serial dilution system in ten consecutive wells, from the initial concentration (1/1) to the highest (1/512). The total broth volume was brought to 200 µL. Microbial inoculum in MH broth as positive control and microbial inoculum in 30% ethanol as negative control were prepared and placed in wells 11 and 12, respectively. For bacteria, the plates were incubated at 37 °C for 24 h, and at 28 °C for 48 h for *Candida*. MIC values were determined as the lowest concentration of the extracts’ dilution that inhibited the growth of the microbial cultures (having the same OD as the negative control), compared to the positive control, as established by a decreased value of absorbance at 450 nm (HiPo MPP-96, Biosan, Latvia). MIC50 was determined as well, representing the MIC value at which ≥50% of the bacterial/yeast cells were inhibited in their growth, considered as the well with the OD value similar to the average between the positive and negative control.

### 4.8. Assessment of Biological Activites

After the phytochemical profile of samples was completed, biological activities were determined in vitro for the 60 min SE. The selected sample presented the highest number of identified compounds and in the highest yields.

#### 4.8.1. Carrageenan Induced Inflammation Model in Rats

An animal model of plantar inflammation on Wistar rats (110–130 g mean weight) was used to evaluate the in vivo anti-inflammatory activity. Acclimatization of the animals was conducted as following: 12 h light/12 h dark cycles, 35% humidity, free access to water, and a normocaloric standard diet (VRF1) and randomized in 4 groups, 8 rats each. Over a course of 4 days, treatment was administered through oral gavage, using a volume of up to 0.25 mL, namely: group 1—carboxymethylcellulose 2% (positive control group—CMC); group 2—Indomethacin 5 mg/body weight (b.w.) in carboxymethylcellulose 1.5% (Indom); group 3—15 mg TPC/b.w./day (60 min SE).

Finally, on the fifth day, inflammation was induced by injecting 100 µL of freshly prepared 1% carrageenan (λ-carrageenan, type IV, Sigma) diluted in normal saline in the right hind footpad [37]. Negative control was established by injecting an identical volume of saline solution in the left hind paw. After the administration of the carrageenan, soft paw tissue was sampled, at 2 h, and 24 h, respectively. The procedure was carried out under an intraperitoneal injection of 90 mg/kg ketamine and 10 mg/kg xylazine. Levels of oxidative stress markers and cytokines were evaluated in the tissue samples after homogenization in 50 mMTRIS–10 mM EDTA buffer (pH 7.4) [37]. Protein content was evaluated in accordance with the Bradford method [38].

#### 4.8.2. Oxidative Stress Assessment

The obtained raw paw tissue homogenates were assessed for malondialdehyde (MDA), reduced glutathione (GSH) and oxidized glutathione (GSSG) levels and GSH/GSSG ratio. Spectrofluorimetry was employed in order to quantify MDA formation, using the 2-thiobarbituric acid method, while the Hu method was used to determine GSH and GSSG levels [39,40].

#### 4.8.3. Proinflammatory Cytokine Assessment

The plantar tissue homogenates were subjected to TNF-α, and IL-6 level evaluation by ELISA assay. The protocol provided by the manufacturer was employed. The results were expressed as pg/mg protein.

#### 4.8.4. Statistical Analysis for the Assessment of Biological Activities

Data were analyzed by one-way ANOVA and Tukey’s multiple comparisons post-test, using GraphPad Prism 8 software. A *p* value < 0.05 was considered statistically significant. The results were expressed as mean ± standard deviation.

## 5. Conclusions

The impact of extraction methods and parameters over the phytochemistry and biological activities of green coffee beans was studied. The highest bioactive compound yields were reached by Soxhlet extraction and ultrasound-assisted extraction. Bioactive compound yields were observed to increase proportionately with the increase in extraction parameters, such as extraction time, temperature, or homogenization speed. Bioactive compound levels were observed to decrease once degradation temperatures were presumably reached. The extracts presented a highly inhibitory effect against Gram-negative bacteria. The 60 min Soxhlet extract, presenting the most favorable results, was eventually selected for in vivo biological activity determination. Non-endogenous antioxidant levels were positively impacted, principally at 2 h after extract administration. Out of the enzymatic antioxidants that were studied, GPx was significantly elevated while the cytokines secretion in the paw tissue was not influenced. In conclusion, aspects such as extraction method and extraction parameters influence both the compositional and biological quality of green coffee beans extracts. In addition, the potential of green coffee beans to serve as a natural, biologically safe, and effective source of bioactive compounds has been demonstrated. This plant material may find practical applications as food supplements, adjuvant therapies, or nutraceuticals as part of the treatment of illnesses of an acute or chronic nature.

## Figures and Tables

**Figure 1 plants-12-00712-f001:**
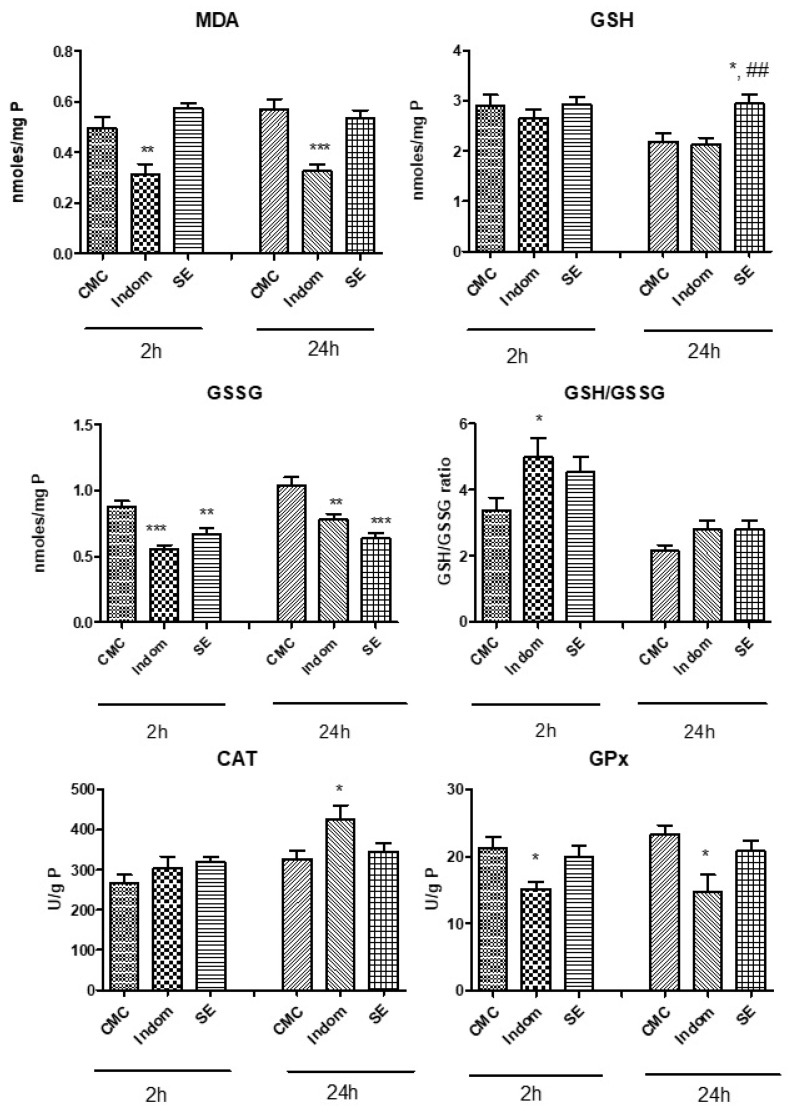
The effect of SE administration on MDA, GSH and GSSG levels, GSH/GSSG ratio, CAT and GPx activities in rat paw tissues at 2 h and 24 h after injection of carrageenan. Values are expressed as means ± SD. Statistical analysis was done by a one-way ANOVA, with Tukey’s multiple comparisons post-test (* *p* < 0.05 and ** *p* < 0.001, *** *p* < 0.0001, all groups vs. control group; ## *p* < 0.01 vs. Indom group).

**Figure 2 plants-12-00712-f002:**
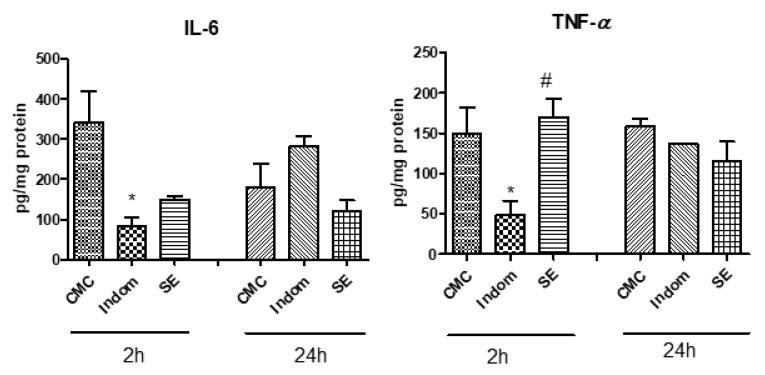
The effect of SE treatment on IL-6 and TNF-α levels in rat paw tissue at 2 h and 24 h after injection of carrageenan. Values are expressed as means ± SD. Statistical analysis was performed using a one-way ANOVA, with Tukey’s multiple comparisons post-test (* *p* < 0.05 vs. control group; # *p* < 0.05 vs. Indom group).

**Table 1 plants-12-00712-t001:** Nomenclature of the evaluated green coffee bean extracts.

Extraction Method	Studied Extraction Parameters				Sample Name
Maceration	*				M
Soxhlet extraction (SE)	Time (min)	20			S20
		40			S40
		60			S60
Turboextraction (TBE)		10 min (2 cycles of 5 min)	Rotation speed (rpm)	4000 6000	T24
					T26
				8000	T28
		20 min (4 cycles of 5 min)		4000	T44
				6000	T46
				8000	T48
Ultrasound-assisted extraction (UAE)		10	Temperature (°C)	30	U13
				40	U14
				50	U15
		20		30	U23
				40	U24
				50	U25
		30		30	U33
				40	U34
				50	U35
Combination of UAE and TBE (UTE)	**				UT

* Parameters remained constant, see Section 4.2.1. Maceration, ** Parameters remained constant, see Section 4.2.5. Combination of UAE and TBE (UTE).

**Table 2 plants-12-00712-t002:** Total polyphenolic content and total flavonoid content of the green coffee beans extracts.

Sample	TPC (GAE mg/mL) *	TFC (QE mM) *
M	1.673 ± 0.061	0.227 ± 0.012
S20	2.008 ± 0.032	0.315 ± 0.002
S40	2.062 ± 0.017	0.225 ± 0.011
S60	2.691 ± 0.026	0.487 ± 0.006
T24	1.375 ± 0.037	0.306 ± 0.016
T26	1.329 ± 0.035	0.161 ± 0.007
T28	1.417 ± 0.037	0.236 ± 0.001
T44	1.493 ± 0.008	0.316 ± 0.002
T46	1.694 ± 0.039	0.302 ± 0.003
T48	1.474 ± 0.028	0.242 ± 0.006
U13	0.443 ± 0.009	0.351 ± 0.012
U14	0.469 ± 0.011	0.245 ± 0.010
U15	0.906 ± 0.047	0.292 ± 0.003
U23	0.720 ± 0.014	0.372 ± 0.001
U24	0.752 ± 0.022	0.288 ± 0.003
U25	1.069 ± 0.005	0.356 ± 0.009
U33	0.676 ± 0.020	0.307 ± 0.009
U34	0.912 ± 0.038	0.373 ± 0.003
U35	1.296 ± 0.061	0.334 ± 0.004
UT	0.918 ± 0.044	0.356 ± 0.003

* Concentrations were expressed as mean ± SD. TPC—total polyphenolic content; GAE mg/mL—mg gallic acid equivalents per mL; TFC—total flavonoid content; QE mM—mM quercetin equivalents.

**Table 3 plants-12-00712-t003:** In vitro antioxidant capacity of the extracts.

Sample	DPPH (TE mg/mL) *	FRAP (TE mM) *	ABTS^+^ (TE mM) *
M	2.610 ± 0.154	17.030 ± 0.959	9.381 ± 1.261
S20	6.109 ± 0.345	22.648 ± 0.354	11.907 ± 0.394
S40	6.154 ± 0.198	20.675 ± 0.357	11.843 ± 0.477
S60	5.876 ± 0.057	23.620 ± 0.510	16.136 ± 0.868
T24	6.192 ± 0.170	18.414 ± 0.479	10.328 ± 0.665
T26	6.140 ± 0.075	20.114 ± 1.347	10.960 ± 1.218
T28	6.123 ± 0.092	21.181 ± 0.543	10.960 ± 0.895
T44	6.074 ± 0.087	20.944 ± 0.830	13.359 ± 0.717
T46	6.128 ± 0.019	22.090 ± 0.674	12.159 ± 0.568
T48	6.116 ± 0.029	19.126 ± 0.725	12.854 ± 0.934
U13	3.796 ± 0.085	17.505 ± 0.428	4.331 ± 0.219
U14	3.757 ± 0.045	17.940 ± 0.181	3.826 ± 0.189
U15	6.425 ± 0.017	23.395 ± 0.585	9.318 ± 0.189
U23	4.737 ± 0.017	22.446 ± 0.068	7.361 ± 1.588
U24	5.639 ± 0.061	18.197 ± 0.332	8.308 ± 0.394
U25	6.336 ± 0.061	13.907 ± 0.181	10.202 ± 0.289
U33	2.653 ± 0.088	16.951 ± 0.49f4	8.056 ± 0.219
U34	4.476 ± 0.029	15.330 ± 0.137	9.760 ± 0.477
U35	9.160 ± 0.041	26.676 ± 0.342	12.475 ± 0.477
UT	8.256 ± 0.174	20.074 ± 0.298	6.604 ± 0.219

* Concentrations were expressed as mean ± SD. DPPH—(2,2-Diphenyl-1-(2,4,6-trinitrophenyl)hydrazyl); TE mg/mL—mg Trolox equivalents per mL extract; FRAP—Ferric-Reducing Antioxidant Power; TE mM—mM Trolox equivalents; ABTS^+^—diammonium 2,2′-azino-bis(3-ethylbenzothiazoline-6-sulfonate; mM TE—mM Trolox equivalents.

**Table 4 plants-12-00712-t004:** Bioactive compounds in the selected green coffee bean extracts (concentrations of polyphenolic and flavonoid compounds were expressed as µg/mL extract, concentrations of sterolic compounds were expressed as ng/mL extract).

Sample	Polyphenolic Compounds	Flavonoid Compounds	Sterolic Compounds					
	Chlorogenic Acid (µg/mL Extract) *	Kaempferol (µg/mL Extract) *	α-Tocopherol (ng/mL Extract) *	γ-Tocopherol (ng/mL Extract) *	Ergosterol (ng/mL Extract) *	Stigmasterol (ng/mL Extract) *	β-Sitosterol (ng/mL Extract) *	Campesterol (ng/mL Extract) *
M	958.49 ± 67.094	1.48 ± 0.073850	<LOQ	20.29 ± 1.826	0.00 ± 0.000	6076.74 ± 668.441	18,609.88 ± 1488.791	993.48 ± 39.739
S20	1321.54 ± 145.369	<LOQ	<LOQ	69.93 ± 4.196	171.49 ± 10.290	5526.52 ± 165.796	27,845.77 ± 1392.288	1424.72 ± 185.214
S40	1376.50 ± 206.474	<LOQ	<LOQ	55.85 ± 2.793	0.00 ± 0.000	3931.22 ± 235.873	20,407.81 ± 612.234	1019.58 ± 142.741
S60	1657.18 ± 215.433	<LOQ	<LOQ	77.98 ± 8.578	50.52 ± 5.557	6030.87 ± 542.778	29,708.84 ± 3862.150	1692.13 ± 135.370
T24	1079.35 ± 43.174	<LOQ	<LOQ	45.90 ± 2.295	0.00 ± 0.000	6497.25 ± 649.726	30,876.08 ± 1852.565	1587.28 ± 142.855
T44	1354.60 ± 176.098	<LOQ	<LOQ	70.14 ± 9.820	133.08 ± 17.300	4829.17 ± 482.917	23,076.85 ± 923.074	1224.94 ± 48.998
T46	1426.62 ± 114.130	<LOQ	<LOQ	32.55 ± 4.232	70.82 ± 2.833	2276.65 ± 136.599	10,381.84 ± 726.729	580.06 ± 34.804
U23	889.94 ± 35.598	<LOQ	59.75 ± 4.183	69.76 ± 6.278	0.00 ± 0.000	862.98 ± 94.927	825.96 ± 66.077	40.20 ± 4.422
U34	1072.11 ± 117.932	<LOQ	61.18 ± 9.178	98.44 ± 3.938	95.01 ± 9.501	5726.65 ± 286.332	32,034.60 ± 4164.498	1471.29 ± 161.842
U35	1270.13 ± 152.415	<LOQ	60.18 ± 7.222	114.79 ± 6.887	129.00 ± 15.479	6296.37 ± 629.637	27,051.26 ± 2164.000	1587.03 ± 95.222
UT	996.31 ± 79.705	<LOQ	61.94 ± 4.955	114.52 ± 12.597	126.41 ± 10.113	5637.12 ± 169.114	28,120.24 ± 2812.024	1312.61 ± 196.892

* Concentrations were expressed as mean ± SD; <LOQ below limit of quantification.

**Table 5 plants-12-00712-t005:** Results for the in vitro qualitative study of antimicrobial activity (disk diffusion test).

Strain		Diameter of Inhibition Area *			
		U35	S60	Amoxicillin	Ketoconazole
Gram-positive	*Staphylococcus aureus* ATCC 6538P	7.15	5.83	24.38	-
	*Enterococcus faecalis* ATCC 29212	6.21	6.18	16.8	-
	*Listeria monocytogenes* ATCC 13932	5.98	5.86	18.96	-
	*Bacillus cereus* ATCC 11778	7.29	6.44	8.83	-
Gram-negative	*Escherichia coli* ATCC 10536	13.38	11.74	13.72	-
	*Salmonella enteritidis* ATCC 13076	13.02	9.85	18.43	-
	*Pseudomonas aeruginosa* ATCC 27853	11.23	10.09	R	-
Fungal	*Candida albicans* ATCC 10231	7.21	6.94	-	23.74

* Inhibition area diameter in mm; R—resistant; U35—extract obtained by ultrasound-assisted extraction at 30 min extraction time and 50 °C extraction temperature; S60—Soxhlet extract obtained at a 60 min extraction time.

**Table 6 plants-12-00712-t006:** Results for the in vitro quantitative study of antimicrobial activity (MIC test).

Strain		U35		S60	
		MIC 100 *	MIC 50 *	MIC 100 *	MIC 50 *
Gram-positive	*Staphylococcus aureus* ATCC 6538P	1/16	1/32	1/16	1/32
	*Enterococcus faecalis* ATCC 29212	1/32	1/64	1/32	1/64
	*Listeria monocytogenes* ATCC 13932	1/16	1/32	1/16	1/32
	*Bacillus cereus* ATCC 11778	1/16	1/16	1/16	1/32
Gram-negative	*Escherichia coli* ATCC 10536	1/8	1/16	1/8	1/16
	*Salmonella enteritidis* ATCC 13076	1/8	1/16	1/8	1/16
	*Pseudomonas aeruginosa* ATCC 27853	1/16	1/16	1/16	1/16
Fungal	*Candida albicans* 10231	1/16	1/16	1/32	1/32

* MIC—minimum inhibitory concentration; MIC 50—minimum concentration required to inhibit 50% of cellular growth; minimum concentration required to inhibit 100% of cellular growth; U35—extract obtained by ultrasound-assisted extraction at 30 min extraction time and 50 °C extraction temperature; S60—Soxhlet extract obtained at a 60 min extraction time.

## Data Availability

Not applicable.

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
