# Peer review of "The Effect of Extraction Methods on Phytochemicals and Biological Activities of Green Coffee Beans Extracts"

_plants, 2023, doi:10.3390/plants12040712_

Round 1

Reviewer 1 Report

The manuscript focuses on green coffee beans, which was not mentioned in the abstract or introduction. Neither was an explanation given why green instead of roasted beans were used.

The method section is very clearly written and understandable. However, the result section is so poorly written - no proper introduction of what is being described, which samples are talked about, poor English - that I cannot judge whether the discussion is sound.

A few minor changes, not including the massive editing required for English language:

Line 37: remove “of” after encompassing

Line 38: Change to “species are of present ..”

Line 41: Combine the two side sentences to not sound like bullet points.

Line 46: Change “and so on” to “etc.”

Line 46-48: Add to Line 40, when you mentioned which species this study focuses on.

Line 48: Add this paragraph immediately after “in shape.” Without a space.

Line 59: Requires references.

Line 63: Remove thus

Line 65: Split the sentence between emerged and conventional

Line 65-67: Rephrase to convey the meaning of the sentence.

Paragraph Line 62-75 and Line 80-83 needs editing by an English speaker.

Line 86-91: Elaborate on the abbreviations, as they are used for the first time.

Line 92: Elaborate what TPC and TFC mean.

Line 93: Which previous chapter?

Line 93-97 do not make any sense.

Line 249: What time? Extraction time?

Line 408: Add a full-stop after Agilent.

Author Response

The effect of extraction method on phytochemicals and biological activities of green coffee beans extracts

(previously titled: A study on the impact of extraction methods over phytochemical profile and related biological activities of green coffee beans extracts)

Octavia Gligor, Simona Clichici, Remus Moldovan, Dana Muntean, Ana-Maria Vlase, George Cosmin Nadăș, Ioana Adriana Buzura-Matei, Gabriela Adriana Filip, Laurian Vlase, and Gianina Crișan

Esteemed Editor and Reviewers,

The authors of this paper kindly wish to state their gratitude for your feedback. Please find listed below the authors’ comments to the current review report.

Reviewer #1 – Comments and replies

Line

Reviewer’s comments

Authors’ comments

The manuscript focuses on green coffee beans, which was not mentioned in the abstract or introduction. Neither was an explanation given why green instead of roasted beans were used.

The term “green coffee beans” has been mentioned in Abstract, line 18, and in the Introduction, in line 40 (the exact plant species is specified), line 56 (reports of this plant material’s biological activities), and line 76 (emphasising the reduced amount of scientific literature available regarding such types of studies).

The reason for which green coffee beans were selected instead of roasted beens is explained in Introduction, lines 51-54: “However, roasting has been noted to promote the degradation of polysaccharides, li-pids, chlorogenic acids, other polyphenolic compounds, trigonelline (a compound also responsible for flavor), and protein denaturation. It has been observed that these changes lead to reduction of the subsequent biological activities of the plant material [4–6].”

The method section is very clearly written and understandable. However, the result section is so poorly written - no proper introduction of what is being described, which samples are talked about, poor English - that I cannot judge whether the discussion is sound.

The authors propose to add a new table the (current Table 1) which contains the nomenclature procedure for the extraction methods as well as the obtained samples of the study. This addition was done with the intention of offering the readers a better understanding of the study.

The abbreviations were explained.

The English was corrected.

37

remove “of” after encompassing

The word was removed.

38

Change to “species are of present ..”

Formulation was changed.

41

Combine the two side sentences to not sound like bullet points.

Sentences were combined.

46

Change “and so on” to “etc.”

Phrase was changed.

46-48

Add to Line 40, when you mentioned which species this study focuses on.

Lines were rearranged: “This article focuses on the species that was first mentioned, Coffea arabica L. The seeds of Coffea arabica L. comprise the majority of the worldwide production of coffee.”

48

Add this paragraph immediately after “in shape.” Without a space.

Space between paragraphs was eliminated.

59

Requires references.

References of the paragraph were reorganized.

63

Remove thus

The word was removed.

65

Split the sentence between emerged and conventional

The sentence was rewritten: “Whereas the terms ‘innovative’ or ‘emerging’ extraction methods, are used for those methods that appeared thereafter.”

65-67

Rephrase to convey the meaning of the sentence.

Sentences were rephrased: “Generally, extraction methods are divided in two main categories, based on the era in which they appeared. Names such as ‘conventional’ or ‘classical’ methods are at-tributed to the methods used prior to the end of the 20th century. Whereas the terms ‘innovative’ or ‘emerging’ extraction methods, are used for those methods that appeared thereafter.”

62-75, 80-83

Paragraph Line 62-75 and Line 80-83 needs editing by an English speaker.

The indicated paragraphs were edited by an English speaker:

“In order to ensure the availability of such bioactive compounds, along with offering high recovery rates from any plant matrix, the extraction process holds much im-portance. Generally, extraction methods are divided in two main categories, based on the era in which they appeared. Names such as ‘conventional’ or ‘classical’ methods are attributed to the methods used prior to the end of the 20th century. Whereas the terms ‘innovative’ or ‘emerging’ extraction methods, are used for those methods that appeared thereafter. Conventional methods entail high temperatures, long periods of time in order for the extraction process to reach completion, large solvent volumes, large amounts of plant material, as well as often hazardous solvents. Maceration, decoction, Soxhlet extraction constitute examples of such conventional methods. These characteristics are presently regarded as unfavorable, thus urging for safer, less time-consuming, alternative means of bioactive compound extraction methods to be discovered, with lower amounts of solvent and plant matrix also being required. Thus, bearing the name ‘green extraction methods’ due to their indicative safety, micro-wave-assisted extraction, ultrasound-assisted extraction, pressurized liquid extraction, etc. are considered clear examples of such methods [11,12].”

And:

“The purpose of this article was to provide a better understanding of the relation-ship between extraction type and extraction conditions and the final extraction yields, as well as to determine the influence such extraction parameters might hold over the biological properties of the obtained green coffee bean extracts, such as antimicrobial activity, antioxidant, and anti-inflammatory effects.”

86-91

Elaborate on the abbreviations, as they are used for the first time.

The authors propose to add a new table the (current Table 1) which contains the nomenclature procedure for the extraction methods as well as the obtained samples of the study. This addition was done with the intention of offering the readers a better understanding of the study.

Abbreviations were explained.

92

Elaborate what TPC and TFC mean.

Abbreviations were explained: “Total polyphenolic content (TPC) and total flavonoid content (TFC)…”

93

Which previous chapter?

Phrase was corrected: “…following the methods detailed in chapter 4. Materials and Methods.”

93-97

Lines do not make any sense.

Lines were rewritten: “Chromatographic evaluation followed only for the samples which presented the highest values for the analyses previously stated. The final step of the study consisted of the biological activity assessment of the samples containing the highest yields of bioactive compounds.”

249

What time? Extraction time?

Line was rewritten: “…with the increase in extraction time…”

408

Add a full-stop after Agilent.

Modification was added.

Reviewer 2 Report

Replace title with “The effect of extraction method on phytochemicals and biological activities of green coffee beans extracts”

Correct linguistic and grammatical errors in the manuscript

Line 20 add such as after “innovative methods”

Line 21, total content of what?

Lines 26, 27, Soxhlet and ultrasound assisted extraction methods give the highest yields of phytochemicals in green coffee beans extracts

Abstract need to be rearranged and rewritten and enriched with some data such as HPLC profile main compounds, antioxidant, antimicrobial activity, and some in vivo results

Reduce keywords

Brief the first two paragraphs in introduction

Add more details about the activities of green coffee bean extracts and add some phytochemical compounds detected by HPLC in previous studies

Results: Indicate the abbreviations once appear in the text

Line 94, analysis

Statistical analysis, Clear the samples replications and add post hoc test as LSD and clear that in all Tables

Why the SD is very big

Line 127, four cycles extraction, rewrite all sentence   

Rewrite all tables head for example Table 3. Bioactive compounds (quantity, mg/100g, g/L,….) in the selected extracts

Calculate the yield and concentration of all extracts

Discussion is short add more discussion about all study objectives

Adjust all scientific names and chemical formula in all manuscript

Line 427, (v/v)

Check the outputs of all references

Author Response

The effect of extraction method on phytochemicals and biological activities of green coffee beans extracts

(previously titled: A study on the impact of extraction methods over phytochemical profile and related biological activities of green coffee beans extracts)

Octavia Gligor, Simona Clichici, Remus Moldovan, Dana Muntean, Ana-Maria Vlase, George Cosmin Nadăș, Ioana Adriana Buzura-Matei, Gabriela Adriana Filip, Laurian Vlase, and Gianina Crișan

Esteemed Editor and Reviewers,

The authors of this paper kindly wish to state their gratitude for your feedback. Please find listed below the authors’ comments to the current review report.

Reviewer #2 – Comments and replies

Point

Reviewer’s comments

Authors’ comments

1

Replace title with “The effect of extraction method on phytochemicals and biological activities of green coffee beans extracts”

The title of the article was replaced with the requested version.

2

Correct linguistic and grammatical errors in the manuscript

The manuscript was revised linguistically and grammatically by a native English speaker.

3

Line 20 add such as after “innovative methods”

Addition was made: “…and such innovative methods as turboextraction,”

4

Line 21, total content of what?

Line was rewritten: “Total polyphenolic and flavonoid content, as well as in vitro antioxidant activity of the resulted extracts were spectrophotometrically determined”

5

Lines 26, 27, Soxhlet and ultrasound assisted extraction methods give the highest yields of phytochemicals in green coffee beans extracts

The sentence was corrected: “. The Soxhlet extraction and ultrasound-assisted extraction gave the highest bioactive compound yields…”

6

Abstract need to be rearranged and rewritten and enriched with some data such as HPLC profile main compounds, antioxidant, antimicrobial activity, and some in vivo results

The Abstract was improved with the additionally requested data.

7

Reduce keywords

Keywords were reduced: green coffee beans; innovative extraction methods; oxidative stress reduction; antimicrobial activity; anti-inflammatory activity; ultrasound-assisted extraction.

8

Brief the first two paragraphs in introduction

The first two paragraphs of the Introduction were shortened.

9

Add more details about the activities of green coffee bean extracts and add some phytochemical compounds detected by HPLC in previous studies

The additionally requested details were added in the Discussion chapter of the paper.

10

Results: Indicate the abbreviations once appear in the text

The authors propose to add a new table the (current Table 1) which contains the nomenclature procedure for the extraction methods as well as the obtained samples of the study. This addition was done with the intention of offering the readers a better understanding of the study.

Abbreviations were explained.

11

Line 94, analysis

Sentence was rewritten, as per the request of the first reviewer: “Chromatographic evaluation followed only for the samples which presented the highest values for the analysis processes previously stated.”

12

Statistical analysis, Clear the samples replications and add post hoc test as LSD and clear that in all Tables

Citation: “The least significant difference (LSD) test is used in the context of the analysis of variance, when the F-ratio suggests rejection of the null hypothesis H 0, that is, when the difference between the population means is significant.

This test helps to identify the populations whose means are statistically different. The basic idea of the test is to compare the populations taken in pairs. It is then used to proceed in a one-way or two-way analysis of variance, given that the null hypothesis has already been rejected.” (from https://link.springer.com/referenceworkentry/10.1007/978-0-387-32833-1_226)

In our tables we presented the mean and Stdev of the determinations, this being the common and most relevant way to show the experimental results for either chemical or biological results.

So, the post hoc test is not applicable / not relevant for such data structure (see similar data presentation structure in other papers recently publish by our research group last year:

https://doi.org/10.3390/antiox12010091

https://doi.org/10.3390/agronomy12102397

https://doi.org/10.3390/pathogens11091065

https://doi.org/10.3390/antiox11020218

In addition, the authors would like to mention that point 4.8.4. Statistical analysis was renamed to 4.8.4. Statistical analysis for the assessment of biological activities, in order to prevent further confusions on this matter.

13

Why the SD is very big

The SD reflects the variability - higher in biological determinations, lower on phytochemical determinations.

14

Line 127, four cycles extraction, rewrite all sentence  

Phrase was rewritten: “Results were evidently different for the ABTS+ assay. In this case, the Soxhlet sample with the 60 min extraction period as well as all TBE samples (of both extraction times, 10 min, and 20 min, respectively) reached the highest levels.”

15

Rewrite all tables head for example Table 3. Bioactive compounds (quantity, mg/100g, g/L,….) in the selected extracts

Each of the tables reunite data sourced from different methodologies. The measurement units are utilized in accordance with the employed method. Therefore, the authors consider that a conversion would be impractical as well as extraneous. Additionally, the title of Table 4, has been expanded to include the measurement units in which the concentration of bioactive compounds was expressed: “Table 43. Bioactive compounds in the selected green coffee bean extracts (concentrations of polyphenolic and flavonoid compounds were expressed as µg/mL extract, concentrations of sterolic compounds were expressed as ng/mL extract)”

16

Calculate the yield and concentration of all extracts

The absolute recovery percentage (yield) is not relevant for our scientific approach as we analysed different chemical classes of bioactive compound with so big differences on hydrophilic versus lipophilic properties (for example we have a higher extraction yield for polyphenolic carboxylic acids in water, flavonoids in water/ethanol or sterolic compounds in hexane). But in our case we wanted not to have focus on yield but on the overall optimal conditions that gives one unitary extract with optimal biological effects/properties.

17

Discussion is short add more discussion about all study objectives

The Discussion chapter was expanded as requested.

18

Adjust all scientific names and chemical formula in all manuscript

Scientific names and chemical formulas were adjusted throughout the text.

19

Line 427, (v/v)

Correction was performed.

Reviewer 3 Report

Green coffee is a product made from unroasted coffee beans. The green coffee bean contains polyphenols, including chlorogenic acid, which have antioxidant and anti-inflammatory properties. In this work, two categories of extraction methods were compared to evaluate the impact of these methods and extraction parameters on the phytochemical and biological activities of green coffee beans. For this purpose, a plantar inflammation model in Wistar rats was used to determine oxidative stress reduction potential and anti-inflammatory activity.

The manuscript is well organized and the subject matter is interesting. The new information is satisfactory and provides useful data for operators in the sector. However, authors need to make some changes to improve the clarity of their article.

Line 92: Enter the full name of the acronyms the first time they appear in the text.

Figures and Tables must be understandable even without reading the text. Supplement the table captions with more information.

Names of bacteria, plants, etc. should be italicised - check all text.

Line 513: Reformulate and integrate the sentence to make it clearer.

Integrate and update references, replacing very old ones.

Author Response

The effect of extraction method on phytochemicals and biological activities of green coffee beans extracts

(previously titled: A study on the impact of extraction methods over phytochemical profile and related biological activities of green coffee beans extracts)

Octavia Gligor, Simona Clichici, Remus Moldovan, Dana Muntean, Ana-Maria Vlase, George Cosmin Nadăș, Ioana Adriana Buzura-Matei, Gabriela Adriana Filip, Laurian Vlase, and Gianina Crișan

Esteemed Editor and Reviewers,

The authors of this paper kindly wish to state their gratitude for your feedback. Please find listed below the authors’ comments to the current review report.

Reviewer #3 – Comments and replies

Point

Reviewer’s comments

Authors’ comments

1

Line 92: Enter the full name of the acronyms the first time they appear in the text.

The authors propose to add a new table the (current Table 1) which contains the nomenclature procedure for the extraction methods as well as the obtained samples of the study. This addition was done with the intention of offering the readers a better understanding of the study.

Abbreviations were explained.

2

Figures and Tables must be understandable even without reading the text. Supplement the table captions with more information.

Captions were supplemented with more information.

3

Names of bacteria, plants, etc. should be italicised - check all text.

Scientific names were italicized.

4

Line 513: Reformulate and integrate the sentence to make it clearer.

The sentence was clarified.

5

Integrate and update references, replacing very old ones.

Old references were removed. Other old references were maintained due to their necessity for study realization and documentation. Another reason for this was the lack of other new scientific data on the subject.

Reviewer 4 Report

The authors evaluate the effect of different extraction methods on the phytochemical composition and on some biological activities of green coffee beans. Although the article is well organized and the results agree with the contents, the level of novelty of this manuscript is quite limited. The relationship between yields and extraction methods under different conditions is already well established. Perhaps it would have been interesting to compare the effect of different extraction solvents.

Regarding the manuscript, I would like to mention some aspects that I think need to be improved.

- The authors describe the HPLC-MS analysis but do not include any parameters, chromatograms, or spectral data.

- In Tables 1, 2 and 3, the statistical significance (p value) of these results is not included

- Formal faults and errors: line 93 (…previous chapter), line 367 (270 µ…) …., must be corrected

- I suggest the convenience of a review in English, especially in the Material and Methods section. Likewise, the wording of some paragraphs, such as the conclusions section, should be improved to allow their content to be clarified.

Author Response

The effect of extraction method on phytochemicals and biological activities of green coffee beans extracts

(previously titled: A study on the impact of extraction methods over phytochemical profile and related biological activities of green coffee beans extracts)

Octavia Gligor, Simona Clichici, Remus Moldovan, Dana Muntean, Ana-Maria Vlase, George Cosmin Nadăș, Ioana Adriana Buzura-Matei, Gabriela Adriana Filip, Laurian Vlase, and Gianina Crișan

Esteemed Editor and Reviewers,

The authors of this paper kindly wish to state their gratitude for your feedback. Please find listed below the authors’ comments to the current review report.

Reviewer #4 – Comments and replies

Point

Reviewer’s comments

Authors’ comments

1

The authors describe the HPLC-MS analysis but do not include any parameters, chromatograms, or spectral data.

Chromatograms, parameters, and spectral data were provided in the Supplementary Material. The authors would prefer to include this information in a separate, Supplementary Material, considering that these data have been already presented in previously published scientific articles, which were cited accordingly in the manuscript, and thus no longer present novelty.

2

In Tables 1, 2 and 3, the statistical significance (p value) of these results is not included

These tables present just the values (mean and Stdev), the comparative statistical analysis for such data structure is not relevant for such data structure.

In addition, the authors would like to mention that point 4.8.4. Statistical analysis was renamed to 4.8.4. Statistical analysis for the assessment of biological activities, in order to prevent further confusions on this matter.

3

Formal faults and errors: line 93 (…previous chapter), line 367 (270 µ…) …., must be corrected

Line 93 was corrected: “…following the methods detailed in chapter 4. Materials and Methods.”

Line 367 was corrected: 270 “µL…”

4

I suggest the convenience of a review in English, especially in the Material and Methods section. Likewise, the wording of some paragraphs, such as the conclusions section, should be improved to allow their content to be clarified.

The manuscript was revised linguistically and grammatically by a native English speaker.

Round 2

Reviewer 1 Report

The manuscript was significantly improved by the authors, but can benefit from another editing with regards to the English language and changes in the discussion to facilitate an understanding of the results. The conclusion section is a very clear statement, whereas the discussion is difficult to follow and to assess what the authors would like to “say”.

Title: methods should be plural

Line 19 extraction singular.

2.4.2 What do 1/16 mean? The point of the result section is to describe the results and point out the highlights for the discussion, instead of referring to a table with an insufficient legend and leaving the reader to figure it out themselves.

2.5. The authors are confusing the relevance of p-values and increase/decrease of antioxidants. E.g. Line 395-397: It is irrelevant whether to state “in a more significant manner” GSH showed no P-value<0.05 for 2h, which means there was no change. However, at 24h SE had a significantly higher (1.5-fold) GSH levels. The biological relevance of this paragraph should be an in- or decrease (only if significant) between SE and the other two treatments. This paragraph needs to be rewritten accordingly.

Line 399: What do you mean by “more important”?

Line 449-455 and 460-478 seems more of an introduction than a discussion to me.

Line 529-531: Change to: …U23), which might indicate these extraction parameters as insufficient.

Please include more statements how the literature described in Line 572-583 is related/agrees with the authors results. It read like a literature report without any connection to the manuscript’s data.

Author Response

The effect of extraction method on phytochemicals and biological activities of green coffee beans extracts

(previously titled: A study on the impact of extraction methods over phytochemical profile and related biological activities of green coffee beans extracts)

Octavia Gligor, Simona Clichici, Remus Moldovan, Dana Muntean, Ana-Maria Vlase, George Cosmin Nadăș, Ioana Adriana Buzura-Matei, Gabriela Adriana Filip, Laurian Vlase, and Gianina Crișan

Esteemed Editor and Reviewers,

The authors of this paper kindly wish to state their gratitude for your feedback. Please find listed below the authors’ comments to the current review report.

Reviewer #1 – Comments and replies

Point

Reviewer’s comments

Authors’ comments

1

The manuscript was significantly improved by the authors, but can benefit from another editing with regards to the English language and changes in the discussion to facilitate an understanding of the results. The conclusion section is a very clear statement, whereas the discussion is difficult to follow and to assess what the authors would like to “say”.

The Discussion chapter was rechecked and additional improvements in phrase formulations were made in order to provide better understanding of the text.

2

Title: methods should be plural

Title was modified: “The effect of extraction methods on phytochemicals and bio-logical activities of green coffee beans extracts”

3

Line 19 extraction singular.

Change was made: “Extraction processes…”.

4

2.4.2 What do 1/16 mean? The point of the result section is to describe the results and point out the highlights for the discussion, instead of referring to a table with an insufficient legend and leaving the reader to figure it out themselves.

In table 6 (section 2.4.2) the antimicrobial potential of the samples was assessed by the MIC method. As explained in the materials and methods section, at point 4.7.2, from the extract’s stock solution (1/1), serial dilutions (2 fold) were performed, until the concentration of 1/512 (1/1, ½, ¼, 1/8, 1/16, 1/32 and so on to 1/512). If the reviewer consider, we can add this in the explanation of the table, but this would repeat the information already presented in the section 4.7.2

5

2.5. The authors are confusing the relevance of p-values and increase/decrease of antioxidants. E.g. Line 395-397: It is irrelevant whether to state “in a more significant manner” GSH showed no P-value<0.05 for 2h, which means there was no change. However, at 24h SE had a significantly higher (1.5-fold) GSH levels. The biological relevance of this paragraph should be an in- or decrease (only if significant) between SE and the other two treatments. This paragraph needs to be rewritten accordingly.

The Results section 2.5. was completely rewritten in accordance with the reviewer's observations.

6

Line 399: What do you mean by “more important”?

The requested correction was made.

7

Line 449-455 and 460-478 seems more of an introduction than a discussion to me.

Other reviewers requested the inclusion of  more discussion about all study objectives in the Discussion chapter, as they considered the chapter too short. The authors believe that reducing the Discussion chapter would contradict these other requests.

8

Line 529-531: Change to: …U23), which might indicate these extraction parameters as insufficient.

The requested change was made.

9

Please include more statements how the literature described in Line 572-583 is related/agrees with the authors results. It read like a literature report without any connection to the manuscript’s data.

The authors have included all recently available scientific data, which are relevant to the study as much as possible.

Reviewer 2 Report

The authors have responded accordingly for all comments and suggestions. The manuscript can be accepted 

Author Response

Thank you for your decision.

Round 3

Reviewer 1 Report

I agree with the authors that you don't need to write the whole method section in the results. However, there should be enough information available in the results section and legend to understand what the authors are trying to convey. Legends for Tabel 5 and 6 need to be rewritten to include details of what data is presented. E.g. Table 5 has an asterisk foot note but no asterisk anywhere in the legend or table.

Author Response

The effect of extraction method on phytochemicals and biological activities of green coffee beans extracts

(previously titled: A study on the impact of extraction methods over phytochemical profile and related biological activities of green coffee beans extracts)

Octavia Gligor, Simona Clichici, Remus Moldovan, Dana Muntean, Ana-Maria Vlase, George Cosmin Nadăș, Ioana Adriana Buzura-Matei, Gabriela Adriana Filip, Laurian Vlase, and Gianina Crișan

Esteemed Editor and Reviewer,

The authors of this paper wish to express their gratitude for your feedback. Please find below the authors’ comments on the current review report.

Reviewer #1 – Comments and replies

Reviewer comments

Authors comments

I agree with the authors that you don't need to write the whole method section in the results. However, there should be enough information available in the results section and legend to understand what the authors are trying to convey. Legends for Tabel 5 and 6 need to be rewritten to include details of what data is presented. E.g. Table 5 has an asterisk foot note but no asterisk anywhere in the legend or table

Legends for Tables 5 and 6 have been rewritten with more details.